

# Hip muscular strength balance is associated with running economy in recreationally-trained endurance runners

Wallace A. Silva[1], Claudio Andre B. de Lira[2], Rodrigo L. Vancini[3] and Marilia S. Andrade[1]

[1] Departamento de Fisiologia, Universidade Federal de São Paulo, São Paulo, São Paulo, Brazil
[2] Faculdade de Educação Física e Dança, Universidade Federal de Goiás, Goiânia, Goiás, Brazil
[3] Centro de Educação Física e Desportos, Universidade Federal do Espírito Santo, Vitória, Espírito Santo, Brazil

## ABSTRACT

**Background**. The percentage of sustained maximal oxygen uptake and the running economy are important factors that determine the running success of endurance athletes. Running economy is defined as the oxygen uptake required to run at a given speed and depends on metabolic, cardiorespiratory, biomechanical, neuromuscular, and anthropometric factors. With regard to anthropometric characteristics, total body mass seems to be a crucial factor for the running economy. Moreover, neuromuscular components, especially knee muscular strength and the strength balance ratio, also seem to be critical for the running economy. In addition to knee muscle strength, hip muscle strength is also an important contributor to running performance on level or hilly ground. However, the relationship between running economy and the hip muscles is unknown. Thus the aim of the present study was to verify whether hip flexor and extensor isokinetic peak torque, the isokinetic strength balance ratio, total body mass and fat free mass were associated with running economy in both sexes and to compare sex differences in physical fitness and isokinetic strength characteristics.

**Methods**. A total of 24 male ($31.0 \pm 7.7$ years, $176.2 \pm 7.3$ cm, and $70.4 \pm 8.4$ kg) and 15 female ($31.3 \pm 6.7$ years, $162.9 \pm 3.9$ cm, and $56.0 \pm 5.3$ kg) recreationally-trained endurance runners were recruited. Maximal oxygen uptake, running economy, conventional (concentric flexors-to-concentric extensors) and functional (concentric flexors-to-eccentric extensors) hip isokinetic strength balance ratios, peak torque of the hip flexor and extensor muscles, total body mass, and fat-free mass were measured. Running economy was assessed on two separate days by means of the energy running cost ($E_c$) using a motorized treadmill at 10.0 and 12.0 km h$^{-1}$ (3% gradient) and 11.0 and 14.0 km h$^{-1}$ (1% gradient).

**Results**. The functional balance ratio was significantly and negatively associated with $E_c$ at 11.0 ($r = -0.43$, $P = 0.04$) and 12.0 km h$^{-1}$ ($r = -0.65$, $P = 0.04$) when using a 3% gradient in male runners. Considering muscular strength, male runners only showed a significant relationship between $E_c$ (assessed at 12 km h$^{-1}$ and a 3% gradient) and peak torque for extensor muscle eccentric action ($r = 0.72$, $P = 0.04$). For female runners, only peak torque relative to total body mass for extensor muscles ($180°$ s$^{-1}$) was positively associated with $E_c$ when assessed at 10 km h$^{-1}$ using a 3% gradient ($r = 0.59$, $P = 0.03$). No significant relationships were found between $E_c$ and total body mass or fat-free mass.

Corresponding authors
Claudio Andre B. de Lira,
andre.claudio@gmail.com
Marilia S. Andrade,
marilia1707@gmail.com

**Discussion**. Given that the functional balance ratio was associated with a better $E_c$, coaches and athletes should consider implementing a specific strengthening program for hip flexor muscles to increase the functional ratio.

# INTRODUCTION

The maximum oxygen-uptake ability, the percentage of sustained maximal oxygen uptake ($\dot{V}O_2$ max), and the running economy are important factors that determine running success in endurance athletes (*Bassett Jr & Howley, 2000*). In addition, while $\dot{V}O_2$ max represents the maximal rate of oxygen uptake, the % $\dot{V}O_2$ max represents the fraction of $\dot{V}O_2$ that a runner can maintain during a running event. Running economy is the metabolic cost required to cover a given distance (*Shaw, Ingham & Folland, 2014*). In this sense, runners with a good running economy experience lower metabolic costs than runners with a poor running economy at the same absolute intensity (*Thomas, Fernhall & Granat, 1999*).

In a 10-km race, almost 65% of the performance variance between runners with a similar $\dot{V}O_2$ max can be explained by running economy, which is measured by $\dot{V}O_2$ (ml kg$^{-1}$ min$^{-1}$) at a common treadmill speed (*Conley & Krahenbuhl, 1980*). Furthermore, running economy depends on metabolic, cardiorespiratory, biomechanical, neuromuscular, and anthropometric factors (*Santos-Concejero et al., 2014*). With regard to the anthropometric characteristics, body composition (fat mass and fat free mass) seems to be a crucial factor. Several studies have identified associations between anthropometric variables and running economy (*Kong & Heer, 2008*; *Dellagrana et al., 2015*; *Hoogkamer, Kram & Arellano, 2017*); for example, *Vernillo et al. (2013)* studied Kenyan marathon runners and concluded that a minimal body fat percentage was desirable, as a higher fat mass correlated with a worse running economy. In addition, total body mass also seems to be important for running performance. *Marc et al. (2014)* studied anthropometric data from the world's top 100 marathon runners between 1990 and 2011 and showed that runners got lighter over the years. In 1990, men weighed an average of 59.6 ± 2.30 kg, while in 2011, they weighed 56.2 ± 1.10 kg. Kenyan athletes also have smaller calf circumferences than boys from other continents (*Larsen, 2003*), which indicates that although muscle mass is primarily responsible for energy use, a reduced muscle mass may be associated with good running economy if there is sufficient muscle mass to provide the required force and support the metabolic rate (*Fletcher & MacIntosh, 2017*). In this manner, the running economy is estimated to worsen when the leg mass increases by 1 kg; however, if the 1 kg increase occurs in the trunk and not the lower limbs, the worsening running economy is less evident (*Jones et al., 1986*).

Muscular strength and endurance also seem to be critical components for the running economy. The endurance of knee flexor and extensor muscles was negatively associated with running economy when assessed by $\dot{V}O_2$ using a fixed treadmill speed (*Westblad, Svedenhag*

*& Rolf, 1996*; *Hayes, French & Thomas, 2011*). Furthermore, there is also evidence to suggest that strength training is able to improve running economy (*Fletcher, Esau & Macintosh, 2009*; *Balsalobre-Fernández, Santos-Concejero & Grivas, 2016*; *Blagrove, Howatson & Hayes, 2017*).

Apart from the importance of muscular strength, the strength balance ratio between agonist and antagonist muscles, which is considered as an index of joint stability, also seems to be of fundamental importance. Two different strength balance ratios have been studied: (1) conventional balance ratio, which is assessed by the concentric peak torque of antagonist muscles divided by the concentric peak torque of agonist muscles and is associated with static joint stability; and (2) functional balance ratio, which is assessed by the eccentric peak torque of antagonist muscles divided by the concentric peak torque of agonist muscles and is associated with dynamic joint stability. *Sundby & Gorelick (2014)* demonstrated that a higher functional balance ratio of the knee joint was associated with a better running economy, and this association was even better than with absolute thigh muscular strength. Higher functional balance ratio values may result in less energy expenditure during running; however, this relationship has been poorly studied for the hip muscles.

When analyzing the kinetics of running, knee flexor and extensors muscles are involved in power generation during the swing and stance phase; however, hip flexor and extensor muscles also contribute significantly to the generation of running power (*Novacheck, 1998*). For example, hip extensor muscles are mainly involved in power generation during the second half of swing, while hip flexor muscles are mainly involved after toe off (*Novacheck, 1998*). Therefore the strength of these muscles also contribute to stride length (*Novacheck, 1998*), which is significantly related to running economy (*Folland et al., 2017*).

Although hip muscle strength is considered to be of fundamental importance to runners, studies regarding the relationship between the hip strength ratio and running economy are scarce. Another interesting point from the running economy literature is that the vast majority of studies only consider level-ground running. Therefore little is known about uphill races, which have become increasingly popular in the last 40 years and seen exponential participation growth (*Hoffman, Ong & Wang, 2010*).

During uphill running, the speed decreases by 0.1–0.3 km h$^{-1}$ for each 1% increase in gradient (*Townshend, Worringham & Stewart, 2010*), which leads to important biomechanical, neuromuscular, and physiological adaptations. Moreover, uphill running is characterized by a shorter swing phase and a greater proportion of the stride cycle spent in the stance position (*Swanson & Caldwell, 2000*). The lower limb muscles also perform higher net mechanical work compared to level running. The increased work demands caused by the hills are met by an increased power output from all joints, particularly the hip (*Vernillo et al., 2017*). Since the hip muscles are significantly involved in hill running, the muscle strength balance between them may also be important for running economy. Therefore it was hypothesized that higher hip muscular strength ratio values imply lower energy expenditure (i.e., a better running economy).

Running economy was typically measured as the oxygen cost of running, which is defined as the oxygen required to cover a given distance (*Foster & Lucia, 2007*; *Ingham et al., 2008*;

*Tam et al., 2012*) or maintain a given speed (*Saunders et al., 2004*). This measurement assumes that the oxygen cost is an index of adenosine triphosphate turnover during submaximal exercise and thus reflects the metabolic cost of running. However, more recent studies have questioned the adequacy of this representation, as the same $\dot{V}O_2$ could reflect different derived energy depending on the substrate (carbohydrate or fat) used (*Fletcher, Esau & Macintosh, 2009*; *Shaw, Ingham & Folland, 2014*). Therefore this study calculated running economy by considering the substrate used, which is likely to provide a more valid (*Fletcher, Esau & Macintosh, 2009*) and reliable (*Blagrove, Howatson & Hayes, 2017*) measurement. It was also measured as the energy cost of running, as previous studies recommended this as the primary measurement of running economy (*Shaw, Ingham & Folland, 2014*; *Shaw et al., 2015*). Moreover, as previous work indicated that the relationship between body mass and oxygen uptake did not increase proportionately (*Lundstrom et al., 2017*), body mass was scaled allometrically and energy running cost ($E_c$) was expressed in kcal kg$^{-0.75}$ km$^{-1}$.

Although a positive relationship does not indicate a cause and effect relationship, the knowledge of a possible association between the hip strength balance ratio and running economy may contribute to a training scheme and therefore lead to improved performance. To the best of the author's knowledge, this is the first study to investigate the association between hip strength balance and $E_c$ (and running economy). Therefore the purpose of this study was to verify whether the hip strength balance ratio, hip muscular strength, total body mass and fat free mass were associated with $E_c$ during flat-floor or uphill running in recreationally-trained endurance runners. Moreover, the study also aimed to compare sex differences in physical fitness and isokinetic strength characteristics.

## MATERIALS & METHODS

### Study design

The physical assessments were divided across 3 days and were separated by 1 week between each visit. The participants continued their regular training program but were asked to refrain from strenuous workouts on the day before each test. The variables evaluated during visit one included $\dot{V}O_2$ max, the ventilatory threshold (VT), the respiratory compensation point (RCP), height, body mass, fat-free mass, and fat mass. During visit two, running economy was evaluated on a motorized treadmill using two running intensities for level-ground (11 and 14 km h$^{-1}$ at a 1% gradient) and two for uphill running (10 and 12 km h$^{-1}$ at a 3% gradient). During visit three, hip flexor and extensor muscle peak torque were evaluated at 60 and 180° s$^{-1}$.

### Subjects

A total of 39 recreationally-trained endurance runners (24 male and 15 female) participated in the study. Male subjects were aged $31.0 \pm 7.7$ years, were $176.2 \pm 7.3$ cm tall, and weighed $70.4 \pm 8.4$ kg. Female subjects were aged $31.3 \pm 6.7$ years, were $162.9 \pm 3.9$ cm tall, and weighed $56.0 \pm 5.3$ kg. Male and female athletes trained $5.0 \pm 1.9$ and $4.1 \pm 1.8$ days/week, respectively, and the training volume was $68.2 \pm 39.8$ and $48.5 \pm 28.6$ km/week for male and female runners, respectively. None of the runners were involved in any other sports.

All participants were informed of the intent and procedures of the study and signed an informed consent form before data collection. The study protocol was approved by the Human Research Ethics Committee of the Federal University of São Paulo and conformed to the principles outlined in the Declaration of Helsinki (CAAE: 50127315.3.0000.5505, UNIFESP, São Paulo, SP, Brazil).

All participants answered the Physical Activity Readiness Questionnaire (PAR-Q) before testing. The inclusion criteria were responding "no" to all PAR-Q questions, being involved in long-distance running for at least one year, training at least four times per week, participating in running events spontaneously, and being able to complete a 10-km race in 45 (men) or 50 (women) min.

The exclusion criteria included suffering from pain or undergoing surgery on the lower limbs within the last six months; presenting metabolic, cardiovascular, neurologic, or endocrine diseases; using medical drugs or any performance-altering substances; or an inability to perform physical activity.

## Procedures

### Anthropometric evaluation

Total body mass (kg), fat-free mass (% and kg), and fat mass (kg) were evaluated using dual-energy X-ray absorptiometry (DPX NT, GE Healthcare, USA). This method was previously shown to be reliable and valid (*Colyer et al., 2016*). Body mass and height were also measured.

### Cardiopulmonary exercise test (CPET)

Maximum incremental treadmill tests on a motorized treadmill (Inbrasport, ATL, Brazil) were performed using a computer-based exercise system (Quark, Cosmed, Italy) with breath-by-breath analysis of ventilatory and metabolic variables. The subjects were instructed to refrain from eating 2 h before the CPET. The gas analyzer was calibrated using gas of a known concentration (16% $O_2$ and 4% $CO_2$; White Martins, Rio de Janeiro, Brazil) and ambient air. The flow meter was calibrated using a 3 L syringe (Hans Rudolph, Inc, Shawnee, KS, USA). All participants were subjected to a 3-min warm-up period at a comfortable running speed, which they selected individually.

The running intensity was increased by 1 km h$^{-1}$ every minute until the subject reached voluntary exhaustion. At the cessation of exercise, participants were asked to rate their dyspnea using the Borg scale (*Borg, 1982*). The initial CPET speed was selected according to the runner's aerobic conditioning (8 or 9 km h$^{-1}$) to guarantee that the total CPET time was between 8 and 12 min (*Buchfuhrer et al., 1983*). This is long enough to provide useful physiological information yet not long enough to burden the patient and staff.

Participants were verbally encouraged to exercise for as long as possible during the CPET. All measured data were fitted as the mean of 20 s. $\dot{V}O_2$ max was always defined as the highest 20-s average $\dot{V}O_2$ value, with the inclusion criteria consistent with conventional guidelines for $\dot{V}O_2$ max (e.g., an inability to sustain the workload, relative heart rate (HR) at maximal exercise >95% age-predicted HR, respiratory exchange rate at maximal exercise >1.1, and $\dot{V}O_2$ plateau (the point at which $\dot{V}O_2$ increases <150 mL min$^{-1}$ for a given increase in workload)) (*Howley, 2007*).

The VT was determined based on an inflection in the ventilation curve, an increase in the ventilatory equivalent for oxygen without an increase in the ventilatory equivalent for carbon dioxide, and an increase in the partial pressure of exhaled oxygen. RCP was determined based on an inflection in the ventilation curve, an increase in the ventilatory equivalent for oxygen and the ventilatory equivalent for carbon dioxide, and a decrease in the partial pressure of exhaled carbon dioxide (*Whipp, Ward & Wasserman, 1986*). The treadmill speed associated with the occurrence of the VT was used as a reference for the individual's maximal moderate exercise capacity. A Suunto Electronics HR monitor (Ambit 2s, Suunto, Finland) was used to record HR in real time.

### Running economy test

Subjects were instructed to arrive at the laboratory in a hydrated state at least 2 h postprandial and to avoid strenuous exercise for 24 h before the test. Running economy evaluation was performed with the same gas analyzer used during the CPET. The test was preceded by a 10-min warm-up period at a comfortable speed of the participant's choice. In the first test, the running speed was set to 11 km h$^{-1}$ (1% gradient) to simulate level-ground running with air resistance (*Jones & Doust, 1996*). In the second test, the running speed was set to 10 km h$^{-1}$ (3% gradient) to simulate uphill running. The test speeds were lower than the VT for all participants, which was confirmed by $\dot{V}O_2$ stabilization during the last 2 min. The criterion used to identify $\dot{V}O_2$ stabilization is an increase $\leq 150$ ml/min during the last 2 min of the test. Each running economy test lasted for 4 min, and subjects had a 5-min rest period (standing fully connected to the metabolic cart) between the two test velocities. In the third test, the running speed was set at 14 km h$^{-1}$ (1% gradient), and in the fourth test, the running speed was set at 12 km h$^{-1}$ (3% gradient) to simulate uphill running. Only participants who presented a VT >14 km h$^{-1}$ participated in the third and fourth tests.

Running economy was assessed by means of E$_c$. For each speed, breath-by-breath measurements were averaged every 10 s during the final minute of each submaximal stage, and $\dot{V}O_2$ and $\dot{V}CO_2$ were averaged to determine the E$_c$ (*Fletcher, Esau & Macintosh, 2009*). The nonprotein respiratory quotient equations (*Péronnet & Massicotte, 1991*) were used to estimate the percentage of fat utilization, and the energy supplied by each substrate was calculated by multiplying the use of fat and carbohydrate by 9.75 and 4.07 kcal, respectively. The E$_c$ of running was chosen to evaluate running economy as it is likely to provide a more valid (*Fletcher, Esau & Macintosh, 2009*) and reliable (*Blagrove, Howatson & Hayes, 2017*) measurement compared to oxygen cost. It is common to report E$_c$ per kg of body mass (*Shaw, Ingham & Folland, 2018*); however, the relationship between body mass and oxygen uptake does not increase proportionately and therefore body mass should be scaled allometrically. As in a previous study, body mass was raised to the power of 0.75 (kcal kg$^{-0.75}$ km$^{-1}$) (*Lundstrom et al., 2017*).

### Isokinetic strength test

The hip strength test was performed on an isokinetic dynamometer (Biodex Medical System, Shirley, NY, USA). A calibration process was conducted on the equipment according to the user manual, which included gravity correction of the lower limb weight. All participants
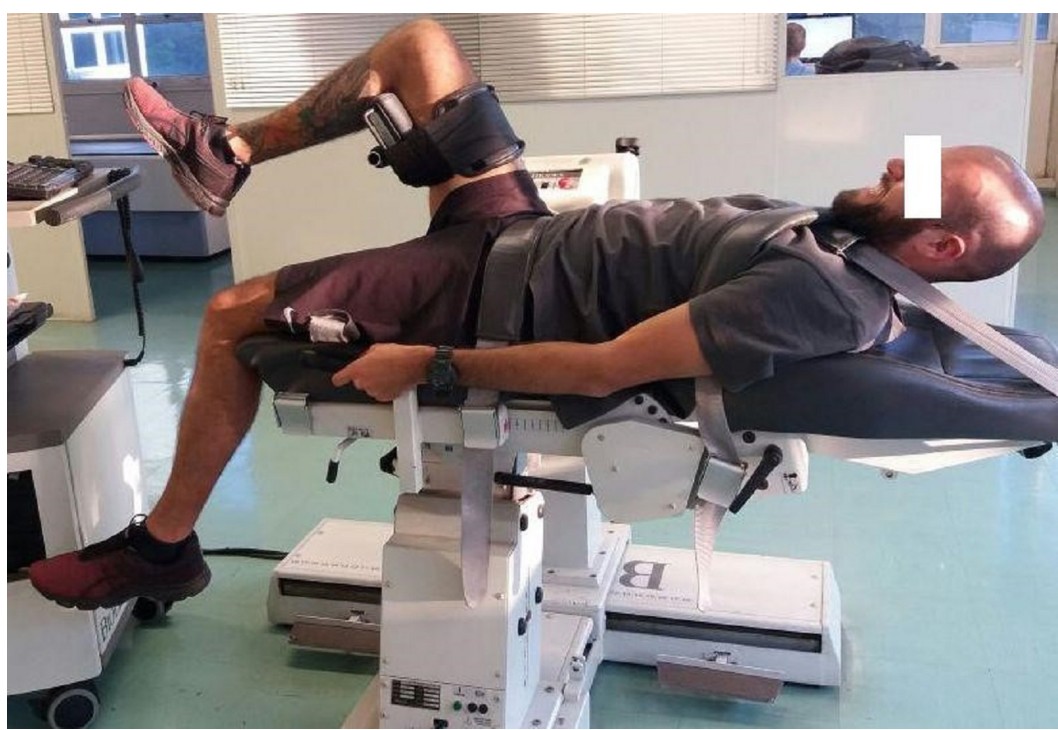

**Figure 1** **Positioning assumed by the participant during isokinetic evaluation of the hip.** Photo credit: Marilia S. Andrade.

performed an initial 4-min warm-up on a motorized treadmill (10–11 km h$^{-1}$). Stretching exercises were avoided as these could influence the strength results (*Mascarin et al., 2015*). Before the test, runners had their trunk and hip stabilized with Velcro straps. According to the manufacturer's specifications, the hip was assessed with the subject lying in a supine position with the seat fully reclined and the knees flexed at 90° (relative to full extension). The hip's axis (greater trochanter used as a bony landmark) stayed in neutral extension (lower limb stayed in a horizontal plane parallel to the ground) and was then aligned with the dynamometer's shaft (Fig. 1). The limb not being tested was held in a knee-flexed and hip-extended position on the table. Hip attachment to the dynamometer was adjusted on the thigh near the knee. The range-of-motion limitations were set beginning from the hip neutrally extended on the table to the hip being maximally flexed. The dynamometer orientation was selected at 0° (relative to the horizontal plane and parallel to the ground).

Concentric hip strength was evaluated during five cycles of maximal flexion and extension at 60 and 180° s$^{-1}$ (*Julia et al., 2010*), while eccentric evaluation was performed during three cycles at 180° s$^{-1}$ (*Zapparoli & Riberto, 2017*). The force–velocity relationship was hyperbolic, and a slow test speed (60°/s) was chosen to evaluate the maximum strength values. On the other hand, functional movements occur at faster angular speeds, therefore a faster test speed (180°/s) was also chosen. Volunteers cannot develop torque at higher angular speeds, therefore the angular speed of 180°/s was the maximum speed used. Prior to the test, all volunteers performed two submaximal contractions to familiarize themselves

**Table 1  General characteristics and running economy parameters of male and female runners.**

| | Male runners (n = 24) | Female runners (n = 15) | P |
|---|---|---|---|
| **Variables** | | | |
| Fat mass (%) | 13.5 ± 6.3 | 21.8 ± 3.9 | <0.01 |
| Fat-free mass (kg) | 57.7 ± 5.8 | 41.7 ± 4.0 | <0.01 |
| $\dot{V}O_2$ max (mL/kg/min) | 61.3 ± 6.0 | 54.5 ± 3.5 | <0.01 |
| 10 km/h uphill $E_c$ (kcal kg$^{-0.75}$ km$^{-1}$) | 3.7 ± 0.2 | 3.8 ± 0.2 | 0.15 |
| 11 km/h flat-floor $E_c$ (kcal kg$^{-0.75}$ km$^{-1}$) | 3.3 ± 0.3 | 3.3 ± 0.2 | 0.95 |
| 12 km/h uphill $E_c$ (kcal kg$^{-0.75}$ km$^{-1}$) (men, n = 9) | 3.5 ± 0.1 | – | – |
| 14 km/h flat-floor $E_c$ (kcal kg$^{-0.75}$ km$^{-1}$) (men, n = 9) | 3.1 ± 0.1 | – | – |

**Notes.**

Data are expressed as mean ± standard deviation.

$E_c$, energy running cost.; $\dot{V}O_2$ max, maximal oxygen uptake.

with the angular speed of the test. Joint evaluation was performed in the dominant lower limb, which was determined by asking the participant which limb they preferred to use to kick a ball. For the purposes of standardization and participant safety, the test started by concentric action (as this was easier to perform) and was followed by eccentric contraction. Peak torque of the hip flexor and extensor muscles were measured, and the conventional strength balance ratio (concentric peak torque of flexor/extensor muscles) and functional strength balance ratio (eccentric peak torque of flexor/concentric peak torque of extensor muscles) were calculated.

## Statistical analysis

All statistical analyses were performed using Statistica software (version 6.0). All data were normally distributed with equal variances (established by the Kolmogorov–Smirnov test), therefore parametric tests were used. Student's t-tests for independent variables were used to compare sex differences relative to anthropometric, CPET, $E_c$, isokinetic testing, and physical training data. Pearson's correlation coefficients were calculated between isokinetic strength variables and $E_c$. Data were expressed as mean ± standard deviation, and the level of significance was set at 0.05.

## RESULTS

The participant characteristics are presented in Table 1. Female runners had a significantly higher fat mass percentage than male runners (Table 1). In addition, male runners had significantly higher fat-free mass (kg) and $\dot{V}O_2$ max levels (Table 1).

The $E_c$ for female runners was not different from male runners when assessed at 10 km h$^{-1}$ during uphill running or 11 km h$^{-1}$ during level-ground running. At the other test speeds (12 km h$^{-1}$ uphill and 14 km h$^{-1}$ flat), the sample was composed only of male runners (n = 9) with a VT >14 km h$^{-1}$ (VT = 14.2 ± 0.4 km h$^{-1}$).

The weekly frequency of training and training volume was not different between the sexes. Hip muscle absolute or relative to total body mass peak torque values in both concentric and eccentric action were higher in male than female runners. The conventional

**Table 2  Peak torque values and conventional and functional balance ratios in the subjects.**

| Variables | Male runners ($n = 24$) | Female runners ($n = 15$) | P |
|---|---|---|---|
| Conventional strength balance ratio | $51.2 \pm 5.1$ | $46.8 \pm 4.3^*$ | 0.04 |
| Functional strength balance ratio | $0.30 \pm 0.08$ | $0.30 \pm 0.12$ | 0.84 |
| Absolute peak torque of extensor muscles at $60° \ s^{-1}$ in concentric action (Nm) | $205.3 \pm 41.9$ | $138.4 \pm 17.4^*$ | <0.01 |
| Peak torque relative to body mass of extensor muscle at $60° \ s^{-1}$ in concentric action (%) | $292.6 \pm 53.2$ | $247.8 \pm 20.2^*$ | <0.01 |
| Absolute peak torque of flexor muscles at $60° \ s^{-1}$ in concentric action (Nm) | $106.9 \pm 19.3$ | $66.3 \pm 10.5^*$ | <0.01 |
| Peak torque relative to body mass of flexor muscle at $60° \ s^{-1}$ in concentric action (%) | $152.5 \pm 23.6$ | $118.7 \pm 15.2^*$ | <0.01 |
| Absolute peak torque of extensor muscles at $180° \ s^{-1}$ in concentric action (Nm) | $172.7 \pm 39.8$ | $115.1 \pm 13.6^*$ | <0.01 |
| Peak torque relative to body mass of extensor muscle at $180° \ s^{-1}$ in concentric action (%) | $245.6 \pm 48.6$ | $206.7 \pm 22.4^*$ | <0.01 |
| Absolute peak torque of flexor muscles at $180° \ s^{-1}$ in concentric action (Nm) | $96.0 \pm 14.2$ | $61.9 \pm 9.1^*$ | <0.01 |
| Peak torque relative to body mass of flexor muscle at $180° \ s^{-1}$ in concentric action (%) | $137.4 \pm 19.2$ | $110.7 \pm 10.2^*$ | <0.01 |
| Absolute peak torque of extensor muscles at $180° \ s^{-1}$ in eccentric action (Nm) | $331.6 \pm 77.3$ | $221.8 \pm 51.0^*$ | <0.01 |
| Peak torque relative to body mass of extensor muscles at $180° \ s^{-1}$ in eccentric action (Nm) | $475.3 \pm 115.5$ | $398.9 \pm 83.9^*$ | 0.03 |
| Absolute peak torque of flexor muscles at $180° \ s^{-1}$ in eccentric action (Nm) | $204.9 \pm 86.6$ | $120.1 \pm 40.1^*$ | <0.01 |
| Peak torque relative to body mass of flexor muscles at $180° \ s^{-1}$ in eccentric action (Nm) | $296.8 \pm 135.1$ | $214.1 \pm 61.6^*$ | 0.03 |

**Notes.**
Data are expressed as mean $\pm$ standard deviation.
*$P < 0.05$ compared to male runners.

balance ratio was also higher in male runners, but the functional balance ratio was not different between the sexes (Table 2).

Regarding the associations between $E_c$ and the strength balance ratios (male runners), correlation coefficients ranged from 0.01 to 0.29 and $-0.72$ to 0.09 for the conventional balance ratio and functional balance ratio, respectively (Table 3). The functional balance ratio was significantly and negatively associated with $E_c$ when assessed at 11 km h$^{-1}$ using a 1% gradient ($r = -0.43$, $P = 0.04$) and at 12 km h$^{-1}$ using a 3% gradient ($r = -0.72$, $P = 0.04$) in male runners (Table 3). For female runners, the associations between $E_c$ and the conventional and functional balance ratio ranged from $-0.16$ to 0.35; however, neither of these associations were significant (Table 4).

Absolute strength values seemed to be unrelated to $E_c$ in both sexes. For male runners, only peak torque for extensor muscles in eccentric action was positively associated with $E_c$ when assessed at 12 km h$^{-1}$ using a 3% gradient ($r = 0.72$, $P = 0.04$). For female runners, only peak torque relative to total body mass for extensor muscles ($180° \ s^{-1}$) in concentric action was positively associated with Ec when assessed at 10 km h$^{-1}$ using a 3% gradient ($r = 0.59$, $P = 0.03$). Finally, total body mass and fat free mass were not significantly associated with $E_c$ in either the male (Table 3) or female (Table 4) runners.

## DISCUSSION

The primary aim of this study was to verify the relationships of isokinetic hip strength, muscular strength balance ratios, total body mass and fat free mass with $E_c$. The main outcome was that the hip functional balance ratio was strongly associated with $E_c$ at 11

**Table 3  Correlation coefficient ($P$ value) between isokinetic and anthropometric variables and energy running cost ($E_c$) for male runners.**

| | $E_c$ at 10 km h$^{-1}$ with a 3% gradient ($n = 24$) | $E_c$ at 11 km h$^{-1}$ with a 1% gradient ($n = 24$) | $E_c$ at 12 km h$^{-1}$ with a 3% gradient ($n = 9$) | $E_c$ at 14 km h$^{-1}$ with a 1% gradient ($n = 9$) |
|---|---|---|---|---|
| **Isokinetic variables** | | | | |
| Conventional strength balance ratio | 0.01 (0.98) | 0.23 (0.29) | 0.05 (0.90) | 0.29 (0.47) |
| Functional strength balance ratio | −0.29 (0.18) | −0.43 (0.04) | −0.72 (0.04) | 0.09 (0.82) |
| Absolute peak torque of extensor muscles at 60° s$^{-1}$ in concentric action (Nm) | −0.01 (0.97) | −0.01 (0.97) | 0.56 (0.14) | −0.14 (0.73) |
| Peak torque relative to body mass of extensor muscle at 60° s$^{-1}$ in concentric action (%) | −0.16 (0.47) | −0.02 (0.92) | 0.32 (0.43) | −0.14 (0.72) |
| Absolute peak torque of flexor muscles at 60° s$^{-1}$ in concentric action (Nm) | 0.01 (0.95) | 0.08 (0.71) | 0.61 (0.10) | 0.01 (0.99) |
| Peak torque relative to body mass of flexor muscle at 60° s$^{-1}$ in concentric action (%) | −0.14 (0.50) | 0.11 (0.60) | 0.31 (0.44) | 0.02 (0.96) |
| Absolute peak torque of extensor muscles at 180° s$^{-1}$ in concentric action (Nm) | 0.16 (0.47) | 0.01 (0.97) | 0.63 (0.09) | −0.19 (0.64) |
| Peak torque relative to body mass of extensor muscle at 180° s$^{-1}$ in concentric action (%) | 0.05 (0.80) | −0.01 (0.97) | 0.49 (0.21) | −0.21 (0.60) |
| Absolute peak torque of flexor muscles at 180° s$^{-1}$ in concentric action (Nm) | 0.10 (0.62) | −0.25 (0.25) | 0.31 (0.45) | 0.02 (0.95) |
| Peak torque relative to body mass of flexor muscle at 180° s$^{-1}$ in concentric action (%) | −0.08 (0.71) | −0.20 (0.37) | −0.10 (0.80) | 0.02 (0.95) |
| Absolute peak torque of extensor muscles at 180° s$^{-1}$ in eccentric action (Nm) | 0.30 (0.16) | 0.25 (0.25) | 0.72 (0.04) | −0.16 (0.69) |
| Peak torque relative to body mass of extensor muscles at 180° s$^{-1}$ in eccentric action (%) | 0.15 (0.70) | 0.60 (0.09) | 0.47 (0.19) | −011 (0.76) |
| Absolute peak torque of flexor muscles at 180° s$^{-1}$ in eccentric action (Nm) | 0.30 (0.16) | 0.35 (0.10) | −0.16 (0.69) | −0.68 (0.06) |
| Peak torque relative to body mass of flexor muscles at 180° s$^{-1}$ in eccentric action (%) | 0.22 (0.57) | 0.68 (0.04) | −0.41 (0.26) | −0.56 (0.11) |
| Total body mass (kg) | 0.18 (0.26) | −0.04 (0.92) | 0.62 (0.10) | 0.05 (0.95) |
| Fat-free mass (g) | 0.27 (0.21) | 0.10 (0.63) | 0.58 (0.13) | −0.02 (0.96) |
| Fat-free mass (%) | −0.11 (0.59) | 0.18 (0.41) | −0.31 (0.44) | 0.23 (0.58) |

(1% gradient) and 12 km h$^{-1}$ (at 3% gradient) in male runners. This indicates that a higher functional balance ratio is associated with better running economy.

The conventional and functional balance ratios reflect the joint stability, therefore it was hypothesized that the higher the ratio, the lower the $\dot{V}O_2$ during running, as the runner would not expend energy for joint stability. *Sundby & Gorelick (2014)* recently aimed to verify the association between running economy and knee stability during running; however, they only studied $E_c$ using a level gradient and only assessed the knee joint in females. The authors found a negative and significant association between the functional balance ratio and running economy, therefore they concluded that runners should consider implementing hamstring exercises to improve their functional balance ratios.

The results from this study showed that the conventional balance ratio was not associated with Ec in either male or female runners. Conversely, the functional balance ratio was

**Table 4  Correlation coefficient ($P$ value) between isokinetic and anthropometric variables and energy running cost ($E_c$) for female runners.**

| | $E_c$ at 10 km h$^{-1}$ with a 3% gradient ($n = 15$) | $E_c$ at 11 km h$^{-1}$ with a 1% gradient ($n = 15$) |
|---|---|---|
| **Isokinetic variables** | | |
| Conventional strength balance ratio | 0.01 (0.98) | 0.35 (0.25) |
| Functional strength balance ratio | −0.02 (0.94) | −0.16 (0.60) |
| Absolute peak torque of extensor muscles at 60° s$^{-1}$ in concentric action (Nm) | −0.17 (0.59) | 0.10 (0.75) |
| Peak torque relative to body mass of extensor muscle at 60° s$^{-1}$ in concentric action (%) | −0.07 (0.80) | 0.04 (0.89) |
| Absolute peak torque of flexor muscles at 60° s$^{-1}$ in concentric action (Nm) | −0.15 (0.62) | 0.36 (0.24) |
| Peak torque relative to body mass of flexor muscle at 60° s$^{-1}$ in concentric action (%) | −0.04 (0.88) | 0.36 (0.24) |
| Absolute peak torque of extensor muscles at 180° s$^{-1}$ in concentric action (Nm) | 0.39 (0.19) | 0.53 (0.07) |
| Peak torque relative to body mass of extensor muscle at 180° s$^{-1}$ in concentric action (%) | 0.59 (0.03) | 0.50 (0.09) |
| Absolute peak torque of flexor muscles at 180° s$^{-1}$ in concentric action (Nm) | −0.28 (0.37) | 0.10 (0.74) |
| Peak torque relative to body mass of flexor muscle at 180° s$^{-1}$ in concentric action (%) | −0.27 (0.39) | 0.06 (0.84) |
| Absolute peak torque of extensor muscles at 180° s$^{-1}$ in eccentric action (Nm) | −0.16 (0.61) | 0.22 (0.48) |
| Peak torque relative to body mass of extensor muscles at 180° s$^{-1}$ in eccentric action (%) | −0.19 (0.53) | 0.21 (0.47) |
| Absolute peak torque of flexor muscles at 180° s$^{-1}$ in eccentric action (Nm) | −0.14 (0.64) | −0.12 (0.69) |
| Peak torque relative to body mass of flexor muscles at 180° s$^{-1}$ in eccentric action (%) | −0.15 (0.60) | −0.11 (0.70) |
| Total body mass (kg) | −0.24 (0.51) | 0.09 (0.71) |
| Fat-free mass (g) | 0.01 (0.98) | 0.13 (0.66) |
| Fat-free mass (%) | 0.38 (0.22) | 0.21 (0.50) |

significantly and negatively associated with $E_c$ when tested at 11 km h$^{-1}$ using a 1% gradient or 12 km h$^{-1}$ using a 3% gradient in male runners. Running speed is a relevant factor in the maintenance of a stable gait, and muscular balance is a great contributor to stability during running (*Sasaki & Neptune, 2006*). Therefore, these data suggest that runners with a higher functional balance ratio between the hip muscles may have a steady and fast running style at submaximal speeds, thus generating less energy cost. However, caution should be taken because when two variables are found to be correlated, it not necessarily shows that one variable causes the other. It is considered a questionable cause logical fallacy when two events occurring together are taken to have established a cause-and-effect relationship. Therefore, the significant and negative relationship between Ec and functional balance ratio, not necessary indicates a cause-and-effect relationship between these two variables.

*Sundby & Gorelick (2014)* studied seven well-trained female runners and 11 recreational female runners and identified a significant relationship between knee joint functional balance ratio and $E_c$ in both groups, thus suggesting that knee muscular strength balance may be important for running economy.

A running task is a sequence of concentric and eccentric work, thus the elastic energy provided by the stretching muscle is used in the subsequent concentric muscle action and results in increased strength production with reduced energy expenditure (*Spurrs, Murphy & Watsford, 2003*). This may help explain the significant relationship between the higher functional balance ratio and $E_c$ that was identified in this study, at least for men. Complementing this data, *Farley et al. (1991)* explained that the metabolic rate increases

at lower speeds as the body does not behave in an optimal spring-like manner and the elastic energy is dissipated as heat. This may provide an explanation for why the functional balance ratio and $E_c$ association was higher at faster running speeds.

On the other hand, absolute and relative peak torque values for hip flexor and extensor muscles presented almost any relationship with $E_c$. Of the 40 strength variables analyzed in the male and female runners, only one was related to $E_c$. *Sundby & Gorelick (2014)* analyzed the relationship between knee flexor and extensor peak torque values and running economy in female runners and obtained similar results. They also found no relationship between muscular strength (absolute or relative to body mass values) and running economy.

Previous studies reported the importance of total body mass in runners (*Joyner, Ruiz & Lucia, 2011*). *Hoogkamer, Kram & Arellano (2017)* stated that approximately 74% of the metabolic cost of running was used to support body mass. They also argued that a reduction in body mass was a good strategy to improve $E_c$. The results from this study showed that total body mass was not associated with $E_c$ in either male or female runners. These results were expected, as $E_c$ was expressed relative to total body mass and therefore the body mass effect on running economy was abolished.

In the same way, the percentage of fat-free mass and absolute values were not related to $E_c$. Although several studies report improved $E_c$ resulting from strength training (*Balsalobre-Fernández, Santos-Concejero & Grivas, 2016*; *Beattie et al., 2017*), caution should be taken with traditional physical training programs, as hip muscle strength values were not related to $E_c$. Moreover, such a method could cause a hypertrophic effect, which may negatively affect $E_c$ (*Beattie et al., 2017*).

**Study limitation**

A number of limitations of the study must be mentioned. First, this study was cross-sectional and we were therefore unable to assess the responsiveness of training over time in analyzed variables. Therefore, future studies should employ a longer period using a longitudinal design. Second, the cross-sectional data made difficult to assess the direction of causality. Nevertheless, we believe that these limitations do not prevent conclusions being drawn from the study.

# CONCLUSION

Highly-trained male and female runners were not different in terms of their running economy. Moreover, the results showed that higher the functional balance ratio is associated with lower the $E_c$. However, absolute values for muscular strength were not related to the running energy cost. Therefore, running economy may be related to greater hip flexor muscle strength relative to extensor muscle strength and not to absolute muscle strength.

# PRACTICAL APPLICATIONS

Considering that the functional balance ratio was significantly and negatively associated with $E_c$, coaches and athletes should consider implementing hip flexor muscle strengthening

exercises to increase the functional balance ratio and improve $E_c$, although the significant correlation between these variables does not imply causation.

### Funding

The authors received no funding for this work.

### Competing Interests

The authors declare there are no competing interests.

### Author Contributions

- Wallace A. Silva conceived and designed the experiments, performed the experiments, analyzed the data, prepared figures and/or tables, authored or reviewed drafts of the paper, approved the final draft.
- Claudio Andre B. de Lira and Rodrigo L. Vancini conceived and designed the experiments, analyzed the data, contributed reagents/materials/analysis tools, prepared figures and/or tables, authored or reviewed drafts of the paper, approved the final draft.
- Marilia S. Andrade conceived and designed the experiments, performed the experiments, analyzed the data, contributed reagents/materials/analysis tools, prepared figures and/or tables, authored or reviewed drafts of the paper, approved the final draft.

### Human Ethics

The following information was supplied relating to ethical approvals (i.e., approving body and any reference numbers):

The study protocol was approved by the Human Research Ethics Committee of the Federal University of São Paulo and conformed to the principles outlined in the Declaration of Helsinki (CAAE: 50127315.3.0000.5505, UNIFESP, São Paulo, SP, Brazil).

### Data Availability

The raw data are provided in Data S1.

### Supplemental Information

Supplemental information for this article can be found online at http://dx.doi.org/10.7717/peerj.5219#supplemental-information.

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
