# Peer review of "Hip muscular strength balance is associated with running economy in recreationally-trained endurance runners"

_PeerJ, doi:10.7717/peerj.5219_

## Round 0.1 · original submission · Major Revisions

Thank you for your submission of this interesting study. The reviewers and I share some significant concerns with your paper. Please respond to each of the reviewers' comments and suggestions, which I believe will greatly improve your paper. Best of luck.

Reviewer 1 ·

Basic reporting

Some rework of the grammar is required. Many ambiguous terms are used throughout the manuscript.

Experimental design

experimental design is relatively sound. As I have elaborated on in my comments however, the choice to express RE in terms of kcal/km is not justified in my opinion.

Validity of the findings

Difficult/impossible to comment on these given the expression of RE itself and some questions surrounding the specifics of how exactly the data were acquired.

Additional comments

General comments:
This study examined the relationships between hip muscular strength and the energy cost of running as well as various body composition parameters on the energy cost of running. The study’s methodologies are reasonably sound; however, I am not convinced that the energy cost of running expressed as kcal/km (ie. in absolute body mass terms) is justified, particularly given the subjects ran at 1% and 3% treadmill gradient. The justification for the expressing Ec in this way was not present in the introduction and so I’m not convinced similar results would be obtained had Ec been expressed in the more usual kcal/kg/km. In and of itself, showing the relationship between Ec (expressed as kcal/km) and body mass is not new () so further justification for the use of Ec expressed in this way is required before the results can be interpreted correctly. See a recent paper by Lundstrom (2017). Further, there is little introductory statements regarding ‘conventional’ and ‘functional’ hip strength ratios or how these are defined. I also have some contention with presenting ‘partial’ results for RE (eg. Only some male subjects could run below AT at 14 km/hr, so the other subjects are (rightfully) excluded). However, this is now a ‘partial’ result of the whole subject cohort. There are also frequent grammatical errors and/or vague statements that require correction and/or clarification. I have elaborated on this and other issues I see throughout the manuscript in my specific comments below; however, given the confusion around how the results were acquired (ie. issues in the methodology) and how these results were interpreted (specifically why Ec is expressed as kcal/km), I cannot confidently comment on the discussion.

Specific comments:
L23 – why introduce the term “running economy (RE)” here when it is not used afterwards? It is simply confusing when energy cost of running is defined and used later on throughout.
L26 – specifically what about ‘body composition’?
L29 – why is hip muscle strength (or strength in general) important for RE?
L35 – define conventional and functional hip isokinetic strength balance ratios
L38 – Level ground running is not 1% slope. I realize this was done to simulate overground running according to Jones and Doust (1996), but they show that this ‘correction’ is only necessary at faster speeds (much faster than the speeds your subjects ran).
L47 – the suggestion that athletes should incorporate strength training into their training plans is not new. See a recent review by Blagrove (Blagrove, Brown, Howatson, & Hayes, 2017; Blagrove, Howatson, & Hayes, 2017).
L72 – ‘the maximal oxygen uptake (VO2max)”…
L75 – VO2max determines the maximal RATE of o2 uptake, not exactly the utmost limit of the aerobic system.
L77 – Bassett & Howley
L79 – the positive relationship between RE and VO2max arises from the inappropriate measurement of RE. This is explained by Fletcher et al. (2009).
L89 – when citing previous literature on any associations with RE, it is important to consider and make obvious to the reader, which studies used RE as the VO2 measured at a common speed vs. those studies which used the more appropriate cost of transport (eg. Kcal/kg/km or J/kg/m etc.).
L91-93 – Fletcher and MacIntosh (2017) offer a much more detailed explanation on the relationship between body mass and energetics of running.
L94-97 – not relevant. Consider removing.
L97 – why is a lower FFM considered to be associated with a good RE? Again, this is already well-explained (at least theoretically) by Fletcher and MacIntosh (2017).
L99 – “neuromuscular components” is quite vague.
L100 – Westerblad et al (1996) SHOWED a negative relationship…
L101-102 – why is hamstring strength, quad eccentric endurance associated with beter running performance?
L103 – I think the evidence is substantial enough to suggest that resistance training is able to improve RE (see review by Blagrove et al. (2017)). However, this is not a result of improvements in any ‘muscular elastic components’ (this statement itself is not physiologically relevant) as shown by strength training interventions to increase tendon stiffness (Fletcher, Esau, & MacIntosh, 2010; Kubo et al., 2010; Kubo, Kanehisa, & Fukunaga, 2002) as well as the minimal energy return of elastic energy from tendons ((Fletcher & MacIntosh, 2015).
L110 – why is the strength ratio important to consider?
L114 – this statement re: speed increases, as in sprinting, is not relevant to the paper, since the measurement of RE is only relevant and measureable below the anaerobic threshold.
L120 – some additional background regarding the relevance to ultra-trail running? RE is more relevant than other distance running events because the intensity of trail running is below the anaerobic threshold. However, Millet (2013) showed ultra-trail runners in fact sacrifice RE, presumably to reduce muscle damage.
L127-128 – vague sentence.
L130 – what defines ‘adequate’?
L131 – cause and effect RELATIONSHIP
L133 - …and therefore LEAD to IMPROVED performance.
L134 – energy cost of running (Ec) is just now being introduced without any definition or how it may differ from RE.
L144-146 – ‘the body composition’… this statement belongs in the intro and needs further elaboration.
L149 – why those specific speeds tested?
L151 – how was ‘well-trained’ defined?
L162 – as the comment above (L151), what is meant by ‘reaching a race pace of….”? That the subjects could sustain that pace for the entire race? If so, then simply state ‘could complete a 10km race in ______ mins”.
L171 – reliability and validity of DEXA to asses fat and fat-free mass?
L183 – replace ‘shortness of breath’ with ‘dyspnea’?
L185 – avoiding substantial dehydration is not the reason graded exercise tests are usually 8-12 minutes.
L190 – superscript ‘1’ in 150 ml min-1
191 – ‘AT was assessed using established criteria’…there are many established criteria for the definition of the anaerobic threshold and how to determine it. A clear definition of how it was determined and whether it was the first or second ventilatory/lactate turnpoint is required. How was the speed at Anaerobic threshold determined specifically?
L203 – as above, the authors need to justify the use of a 1% treadmill gradient, especially when considering the effect of body mass on RE.
L204 – the TESTED speeds WERE lower than the AT… how was this confirmed?
L209 – how many runners were able to run below AT at 14 km/hr?
L210 – how long were the RE tests? How was steady state confirmed in the RE tests specifically?
L213 – to determine the Ec USING RER as proposed by Shaw et al. 2013. Really, it was Fletcher et al. (2009) who proposed this (and who is referenced by Shaw).
L217 – were the left and right legs tested?
L223 – gravity correction of what?
L225 – 90 deg relative to what?
L226 – what is meant by ‘neutral extension’?
L228 – there is no mention of a table with the Biodex. Perhaps a figure would be best to show the specific subject test set up?
L231 – 0 deg. Relative to what?
L232 – replace ‘evaluation’ with ‘strength’
L240-242 – Unless I missed it, what is CR and FR and what does it mean pragmatically?
L247 – replace ‘gender’ with ‘sex’.
L260-265 – again, the authors need to justify why expressing Ec as kcal/km is most appropriate. When expressed relative to body mass, in fact the male subjects are more economical (eg. 1.15 kcal/kg/km for males and 1.21 kcal/kg/km for females when taking mean values for BM).

Refernces:
Blagrove, R. C., Brown, N., Howatson, G., & Hayes, P. R. (2017). Strength and Conditioning Habits of Competitive Distance Runners. Journal of Strength and Conditioning Research, 1. http://doi.org/10.1519/JSC.0000000000002261
Blagrove, R. C., Howatson, G., & Hayes, P. R. (2017). Effects of Strength Training on the Physiological Determinants of Middle- and Long-Distance Running Performance: A Systematic Review. Sports Medicine, 1–33. http://doi.org/10.1007/s40279-017-0835-7
Fletcher, J. R., Esau, S. P., & Macintosh, B. R. (2009). Economy of running: beyond the measurement of oxygen uptake. Journal of Applied Physiology (Bethesda, Md. : 1985), 107(6), 1918–1922.
Fletcher, J. R., Esau, S. P., & MacIntosh, B. R. (2010). Changes in tendon stiffness and running economy in highly trained distance runners. European Journal of Applied Physiology, 110(5), 1037–1046.
Fletcher, J. R., & MacIntosh, B. R. (2015). Achilles tendon strain energy in distance running: consider the muscle energy cost. Journal of Applied Physiology (Bethesda, Md. : 1985), 118(2), 193–9. http://doi.org/10.1152/japplphysiol.00732.2014
Fletcher, J. R., & MacIntosh, B. R. (2017). Running Economy from a Muscle Energetics Perspective. Frontiers in Physiology, 8(June), 1–15. http://doi.org/10.3389/fphys.2017.00433
Jones, A. M., & Doust, J. H. (1996). A 1% treadmill grade most accurately reflects the energetic cost of outdoor running. Journal of Sports Sciences, 14(4), 321–327.
Kubo, K., Kanehisa, H., & Fukunaga, T. (2002). Effects of resistance and stretching training programmes on the viscoelastic properties of human tendon structures in vivo. The Journal of Physiology, 538(Pt 1), 219–226.
Kubo, K., Tabata, T., Ikebukuro, T., Igarashi, K., Yata, H., & Tsunoda, N. (2010). Effects of mechanical properties of muscle and tendon on performance in long distance runners. European Journal of Applied Physiology, 110(3), 507–514. http://doi.org/10.1007/s00421-010-1528-1
Lundstrom, C. J., Biltz, G. R., Snyder, E. M., & Ingraham, S. J. (2017). Allometric scaling of body mass in running economy data: An important consideration in modeling marathon performance. Journal of Human Sport and Exercise, 12(2). http://doi.org/10.14198/jhse.2017.122.03
Millet, G. Y., Hoffman, M. D., Morin, J. B., Millet, G. Y., Hoffman, M. D., & Morin, J. B. (2013). Sacrificing economy to improve running performance −− a reality in the ultramarathon ? Sacrificing economy to improve running performance — a reality in the, (April 2012), 507–509. http://doi.org/10.1152/japplphysiol.00016.2012

Reviewer 2 ·

Basic reporting

The English language is clear.
The introduction was able to present the study problematic.
The paper structure is conforms to journal guidelines.
There is not a supplement file.

Experimental design

Original research within journal scope.
The research question is well defined and relevant.
The Methods section is well described with sufficient detail to replicate the study.

Validity of the findings

Conclusion is well stated, however the conclusions did not support the study results (see general comments).
Many speculations were supplied. The authors highlight these speculations.

Additional comments

The manuscript entitled “Hip muscular strength balance and body composition evaluated by the DXA are associated with running economy in well-trained runners” tests the relationship between running economy and hip flexor and extensor peak torque, strength balance ratio (CR and FR), and body composition. Although the topic is interesting, some concerns should be considered. The following issues need to be addressed.

Title
Suggestion: Hip muscular strength balance and body composition are associated with running economy in recreationally trained runners.

Abstract
Lines 46-47: The authors stated that strengthening programs should improve neural factors instead muscular hypertrophy. How the results of the present study support this statement? (In the conclusion section the reviewer will discuss again this aspect).

Introduction
In general, the introduction is well writing, congratulations.
Lines 100-102: Are there more current references?
Lines 113-116: This statement is more related with sprinters than endurance runners? How hip muscles that are involved in the power generation can influence running during submaximal intensities (i.e., running economy)?

Material and Methods
Line 151: Please change “…recreational, well-trained endurance runners..” to “ recreationally trained endurance runners…”.
Lines 182-183: Please, provide the initial speed of CPET.
Lines 197-210:
Why the authors choose 3% slope for uphill running?
The reviewer main concern is related to running economy velocities. In this study, the inclusion criteria was pace below 4min:30s (i.e., ~13 km/h) for male and 5min:00s (i.e., ~12 km/h). However, considering that trained endurance runners can run 10 km race at least 100% of AT, the reviewer believe that 12 (uphill) and 14 (level ground) km/h are very high speed to running economy, for all runners analyzed. Although nine runners had AT above 14 km/h, these velocities (12 and 14 km/h) are in the heavy domain, which has VO2 slow component. Additionally, for 10 (3% slope) and 11 (1% slope) km/h is possible that some runners also reach the heavy domain, which preclude the running economy analysis.
Line 246: Considering the sample size, the normality test of Shapiro-Wilk is correct.
Why the authors did not use multiple regression analysis?

Results
Lines 260-263: This result was expected. Since female runners has lower VO2MAX than male runners. Thus, the reviewer suggests add this result after “…higher values for VO2MAX…” (Line 258), as a consequence of lower VO2MAX for female runners.
Lines 265- 266: The authors stated “this represents a moderate running intensity (below AT)”, actually below of AT the running intensity can be moderate or heavy. The intensity is moderate when exercise is performed without blood lactate change, that is, the speeds used in the study indicate that running intensity is heavy for most runners.
Suggestion: The authors can show the joint stability classification, that is, how many runners have good or bad joint stability assessed by CR and FR.

Discussion
Lines 316-318: The authors stated “ … the better ratios the runner has, the lower the VO2 should be during running…”. What is better ratios? For example, FR above 1 is good, but if FR value was 1.70, is this good too? The authors need show the PT, CR and FR values (mean and standard deviation).

Lines 324-326: CR and FR showed no relationship with Ec. Perhaps CR and FR values present a large range. Again, is necessary demonstrate CR and FR values.
Line 353: Please, provide the number of participants of cited study.
Lines 382-383: The authors stated “In this regard, strengthening programs, aiming to improve muscular strength as a consequence of neural factors instead of muscular hypertrophy, should be better for Ec”. The current study cannot support this statement.

Conclusion
Regarding neural factors, the authors need be careful with statements. Since just hip muscle strength was assessed. Thus, you cannot stated that resistance training for runners should avoid hypertrophy and focus on neural factors. Perhaps, eccentric contractions and stretching shortening cycle should be the target for endurance runners.

·

Basic reporting

As mentioned in the general comments, the manuscript should be shortened to make it more concise. Also, a thorough review of English language (especially the structure of some sentences) should be made.

Experimental design

See general comments below. Research objectives may need precisions based on between-sex analyses. Overall very well done.

Validity of the findings

Please provide more details on p-values. Also, make sure conclusions are based on findings from this study.

Additional comments

This manuscript describes a study which aimed at cross-sectionally investigating the association between hip muscles and body composition and running economy. This is an interesting topic, but I feel this manuscript needs major revisions before it can be considered for publication. First, I think the authors can justify their study better. Also, the manuscript could be significantly shortened and more focused.

General comments:
I found the introduction of the paper to be lengthy, and sometimes confusing as there are a lot of speculations (the purpose of an introduction is simply to show the state of literature in a concise manner so that it justifies the current study). Better structure and more ‘to-the-point’ arguments would improve readability. Then, the authors interchangeably use the terms ‘recreationally trained’ and ‘well-trained’ – which is it? Overall, the results section is really heavy to read and should be shortened. The impressive number of abbreviations and numbers (a lot of them are repeated in Tables) does not make it easy to read. Finally, the authors talk about ‘stability’ during running; however, ‘stability’ is a very vague term and is not the topic of this study or isn’t a construct that can actually be assessed.

Abstract:
- Please limit the use of abbreviations as it becomes somewhat confusing. For example, peak torque is usually maintained in its complete form and does not need to be changed to PT.
- Conclusions from the abstract are not related to findings of this study. How do you know if strengthening hip flexors will result in better running economy?
- The last sentence needs to be restructured as it doesn’t make any sense.

Introduction
- L90: Restructure sentence
- L129: why would a strength ratio improve RE, since contractions are submaximal during running?
- Better balance between arguments relating to hip muscles, body composition and uphill running as they relate to running economy should be put forward, and more focused.

Methods
- L144-146: This sentence sounds like it should be in the introduction instead.
- L161: could you describe ‘long distance running’? How many km per week to be included?
- L224-231: Please provide a figure to facilitate comprehension. This paragraph could be reduced in length at the same time.
- L239: see previous comment about PT
- Were strength data normalized to body mass?
- L247: use sex instead of gender. Also, I don’t recall seeing a comparison of men and women in the study objectives. If it is an objective, you should clearly state that.

Results
- L255-258: Data is repeated in text and Tables. I suggest taking that out of the text to improve readability and refer to tables only.
- L284: This should be described in Methods.

Discussion
- L303-308: This paragraph looks like a Methods paragraph and does not belong to a Discussion.

Conclusion
- Revise first sentence. Also, your conclusion is too succinct and does not reflect findings from this study very well.

Tables 2-3: Please provide detailed P-values for correlations. I’m not sure to understand why FFM (g) and FFM (%) are opposite. Unless I’m wrong, the higher the FFM mass, the higher the FFM % should be.

---

## Round 0.2 · Minor Revisions

Thank you for your resubmission. I have sent the revised manuscript back to the reviewers who have made a few further comments, mostly for clarity. The reviewers and I would like to commend you for your attention to detail in your responses and revisions. There are, however, just a few more revisions required prior to acceptance. Reviewer 1 has provided a response to a few of your revisions/answers. Please address these on resubmission. In particular, please carefully address the comment under "validity of the findings" in regards to the correlation/causation discussion points. I agree with Reviewer 1 that the discussion should be revised to temper the language of the discussion to soften any suggestions of causation. Thanks again and I look forward to the resubmission.

Scotty

Reviewer 1 ·

Basic reporting

the revised manuscript has improved substantially since its original submission.

Experimental design

the experimental design fills a small, but interesting, knowledge gap. Methodology has improved substantially since first submission.

Validity of the findings

The authors should temper their discussion to reflect that many results are correlated and correlation does not necessarily imply causation.

Additional comments

The authors should be commended for a thorough revision of the submitted manuscript. I think the manuscript is improved substantially from its previous version.

With regards to my specific comments, I have but a few clarifications that should be addressed. I have highlighted my responses to the author responses in bold below.


L26 – specifically what about ‘body composition’?
Answer: Thank you for your comment. Total body mass seems to be a crucial factor for RE.
Thank you for clarifying. I suggest then to replace ‘body composition’ with total body mass throughout the manuscript (eg. line 32 of revised manuscript). Further, remove ‘body composition’ from line 50 (“no significant relationships were found between Ec, today body mass or body composition). Or simply clarify that ‘body composition’ refers to fat mass, and replace body composition with fat mass.

L47 – the suggestion that athletes should incorporate strength training into their training plans is not new. See a recent review by Blagrove (Blagrove, Brown, Howatson, & Hayes, 2017; Blagrove, Howatson, & Hayes, 2017).
Answer: There is evidence that strength training is beneficial to RE (Blagrove et al. 2017; Denadai et al. 2017); however, the results are still contradictory (Blagrove et al. 2017). Several general strengthening programs have been evaluated, including isometric resistance training (Johnston et al. 1997), explosive resistance training (Berryman et al. 2010), plyometric training (Saunders et al. 2006), and a combination of different strengthening programs (Giovanelli et al. 2017). However, the present study does not suggest that athletes should incorporate general strength training into their training plans. The present study aimed to verify if the hip flexor and extensor isokinetic peak torque (PT) and isokinetic strength balance ratio were associated with running Ec. The main finding was that the hip functional strength balance (flexor eccentric-to-extensor concentric) was more strongly associated with Ec than absolute muscular strength. In light of these results, a specific eccentric strengthening program directed to the hip flexor muscles may improve the functional balance ratio and Ec. This sentence has been rewritten for clarity.
Denadai BS, de Aguiar RA, de Lima LC, et al. Explosive training and heavy weight training are effective for improving running economy in endurance athletes: a systematic review and meta-analysis. Sports Med. 2017;47(3):545–54.
Johnston RE, Quinn TJ, Kertzer R, Vroman NB. Strength training in female distance runners: impact on running economy. J Strength Cond Res. 1997;11(4):224–9.
Berryman N, Maurel DB, Bosquet L. Effect of plyometric vs. dynamic weight training on the energy cost of running. J Strength Cond Res. 2010;24(7):1818–25.
Saunders PU, Telford RD, Pyne DB, et al. Short-term plyometric training improves running economy in highly trained middle and long distance runners. J Strength Cond Res. 2006;20(4):947–54.
Giovanelli N, Taboga P, Rejc E, Lazzer S. Effects of strength, explosive and plyometric training on energy cost of running in ultra-endurance athletes. Eur J Sport Sci. 2017;17(7):805–13.
Thank you for a thoughtful and insightful response. The reference to Fletcher et al. 2009 likely refers to Fletcher et al. 2010. Please revise.

L47 – the suggestion that athletes should incorporate strength training into their training plans is not new. See a recent review by Blagrove (Blagrove, Brown, Howatson, & Hayes, 2017; Blagrove, Howatson, & Hayes, 2017).
Answer: There is evidence that strength training is beneficial to RE (Blagrove et al. 2017; Denadai et al. 2017); however, the results are still contradictory (Blagrove et al. 2017). Several general strengthening programs have been evaluated, including isometric resistance training (Johnston et al. 1997), explosive resistance training (Berryman et al. 2010), plyometric training (Saunders et al. 2006), and a combination of different strengthening programs (Giovanelli et al. 2017). However, the present study does not suggest that athletes should incorporate general strength training into their training plans. The present study aimed to verify if the hip flexor and extensor isokinetic peak torque (PT) and isokinetic strength balance ratio were associated with running Ec. The main finding was that the hip functional strength balance (flexor eccentric-to-extensor concentric) was more strongly associated with Ec than absolute muscular strength. In light of these results, a specific eccentric strengthening program directed to the hip flexor muscles may improve the functional balance ratio and Ec. This sentence has been rewritten for clarity.
Denadai BS, de Aguiar RA, de Lima LC, et al. Explosive training and heavy weight training are effective for improving running economy in endurance athletes: a systematic review and meta-analysis. Sports Med. 2017;47(3):545–54.
Johnston RE, Quinn TJ, Kertzer R, Vroman NB. Strength training in female distance runners: impact on running economy. J Strength Cond Res. 1997;11(4):224–9.
Berryman N, Maurel DB, Bosquet L. Effect of plyometric vs. dynamic weight training on the energy cost of running. J Strength Cond Res. 2010;24(7):1818–25.
Saunders PU, Telford RD, Pyne DB, et al. Short-term plyometric training improves running economy in highly trained middle and long distance runners. J Strength Cond Res. 2006;20(4):947–54.
Giovanelli N, Taboga P, Rejc E, Lazzer S. Effects of strength, explosive and plyometric training on energy cost of running in ultra-endurance athletes. Eur J Sport Sci. 2017;17(7):805–13.
Thank you for a thoughtful and insightful response. The reference to Fletcher et al. 2009 likely refers to Fletcher et al. 2010. Please revise.

L149 – why those specific speeds tested?
Answer: The force–velocity relationship was hyperbolic, and a slow test speed (60º/s) was chosen to evaluate the maximum strength values. On the other hand, functional movements occur at faster angular speeds, therefore a faster test speed (180º/s) was also chosen. Volunteers cannot develop torque at higher angular speeds, therefore the speed of 180º/s was the maximum speed used. These tests speeds were previously used by Julia et al. (2010) and Zapparoli and Riberto (2017).
Thank you for elaborating. The information above should be in the revised manuscript to clarify why 60 and 180º/s were specifically chosen.

L204 – the TESTED speeds WERE lower than the AT… how was this confirmed?
Answer: The grammatical errors have been corrected.
The CPET test was performed to individually check the velocity corresponding to the VT. After verifying the speed corresponding to the VT in each volunteer, we chose a speed below the minimum VT (11 km.h-1) to evaluate the RE. The RE at 14 km.h-1 was only evaluated in a small subgroup of athletes (n = 9) who presented VT at 15 km.h-1. Therefore we can confirm that all volunteers ran at a moderate domain during the RE test using a 1% gradient.
When the RE test was performed using a 3% gradient, the treadmill speed was decreased by 1 or 2 km/h (from 11 to 10 km.h-1 and from 14 to 12 km.h-1). Toownshend, Worringham & Stewart (2010) demonstrated that the speed decreased by 0.1–0.3 km/h for each 1% gradient increase during uphill running. Considering that the treadmill grade increased by 2% (from 1% to 3%) in the RE test, the speed need to be reduced by at least 0.6 km.h-1. As the treadmill speed was reduced by 1 or 2 km.h-1, we conclude that the test was also performed in moderate phase.
Townshend AD, Worringham CJ, Stewart IB. 2010. Spontaneous pacing during overground hill running. Medicine and Science in Sports and Exercise 42:160–169

The revised methodology is substantially improved; however, specifically, the revisions do not address how ‘VO2 stabilization’ was defined (eg. change of <100 ml/min over the final 2 mins).


L210 – how long were the RE tests? How was steady state confirmed in the RE tests specifically?
Answer: RE tests lasted for 4 minutes. The steady state was confirmed by VO2 stabilization during the last 2 minutes of the test. This information has been included in the text for clarification.
The revised methodology is substantially improved; however, specifically, the revisions do not address how ‘VO2 stabilization’ was defined (eg. change of <100 ml/min over the final 2 mins).

Reviewer 2 ·

Basic reporting

The English language was improved.
The introduction was able to present the study problematic.
The paper structure is conforms to journal guidelines.
There is not a supplement file.

Experimental design

Original research within journal scope.
The research question is well defined and relevant.
The Methods section is well described with sufficient detail to replicate the study.

Validity of the findings

Conclusion was improved.
Many speculations were supplied. The authors highlight these speculations.

Additional comments

My comments have been addressed. Thank you.

---

## Round 0.3 · accepted · Accept

Thank you for your resubmission and thoroughly addressing the comments. Congratulations on your acceptance!

Scotty